

**The silica-carbon biogeochemical cycle in the Bohai Sea and its responses to the changing terrestrial loadings**

Jun Liu[1,2], Lex Bouwman[3,4], Jiaye Zang[1], Chenying Zhao[1], Xiaochen Liu[3], Xiangbin Ran[1,3]

1. Research Center for Marine Ecology, First Institute of Oceanography, State Oceanic Administration, Qingdao 266061, China

2. Key Laboratory of Marine Chemistry Theory and Technology, Ministry of Education, College of Chemistry and Chemical Engineering, Ocean University of China, Qingdao 266100, China

3. Department of Earth Sciences–Geochemistry, Faculty of Geosciences, Utrecht University, Utrecht, 3508 TA, The Netherlands

4. PBL Netherlands Environmental Assessment Agency, Bilthoven, 3720 AH, The Netherlands

*Correspondence to:* Xiangbin Ran (rxb@fio.org.cn)



**Abstract**

Silicon (Si) and carbon (C) play key roles in the river and marine biogeochemistry.
The Si and C budgets for the Bohai Sea were established on the basis of
measurements at a range of stations and additional data from the literature. The results
show that the spatial distributions of reactive Si and organic C (OC) in the water

column are largely affected by the riverine input, primary production and export to the
Yellow Sea. Biogenic silica (BSi) and total OC in sediments are mainly from marine
primary production. The major supply of dissolved silicate (DSi) comes from benthic
diffusion, riverine input alone accounts for 17% of reactive Si inputs to the Bohai Sea;
the dominant DSi removal from the water column is diatom uptake, followed by

sedimentation. Rivers contribute 47% of exogenous OC inputs to the Bohai Sea; the
dominant outputs of OC are sedimentation and export to the Yellow Sea. The net
burial of BSi and OC represent 3.3% and 1.0% of total primary production,
respectively. Primary production has increased by 10% since 2002 as a result of
increased river loads of DSi and BSi. Our findings underline the critical role of

riverine Si supply in primary production in coastal marine ecosystems.

**Key words:** Bohai Sea; dissolved silicate; biogenic silica; organic carbon; flux and
budget; primary production





## 1. Introduction

Diatoms control a large part of primary production in marine ecosystems, with more than 50% in the global ocean and more than 75% in coastal waters (Nelson et al., 1995; Rousseaux and Gregg, 2014). The consumption of dissolved silicate (DSi) and production of biogenic silicon (BSi) is primarily controlled by primary production by diatoms (Ragueneau et al., 2000; Tréguer and De La Rocha, 2013).


Although ocean margins cover only 8% of the global ocean area (Berner, 1982), the production and accumulation rates of BSi and organic carbon (OC) in these areas are significantly higher than in the open ocean (Hedges and Kiel, 1995; Tréguer and De La Rocha, 2013). Rivers are the dominant Si and OC source in coastal marine

ecosystems, accounting for up to 80% of total exogenous input (Bauer et al., 2013; Regnier et al., 2013; Tréguer and De La Rocha, 2013). However, large parts of the world's coastal marine ecosystems have been changing due to decreasing riverine Si discharge as a result of Si trapping in reservoirs (Conley, 1997; Humborg et al., 1997). The decreasing input of Si may lead to a shift from a system dominated by diatoms to

one dominated by non-siliceous phytoplankton (Humborg et al., 1997, 2000; Tréguer and De La Rocha, 2013; Rousseaux and Gregg, 2015), which may influence the functioning of coastal marine ecosystems, especially with respect to the carbon (C) cycle.

The Bohai Sea is a semi-enclosed, shallow shelf water body of the North-western Pacific Ocean. A large number of rivers drain into the Bohai Sea, typically with densely populated and industrialized coastal areas. Ongoing human activities (dam construction, agriculture and industry) have induced important changes in the river nutrient concentration and composition (Gong et al., 2015; Liu, 2015).


Dam construction has caused decreased Si transport by the Yellow River (Liu, 2015; Ran et al., 2015), and distorted nutrient stoichiometry (Tang et al., 2003; Ning et al., 2010; Liu et al., 2011), which changed primary production and phytoplankton



composition (Tang et al., 2003; Lin et al., 2005; Ning et al., 2010). Phytoplankton

abundance had decreased in the period of 1959–1999 (Tang et al., 2003) and dominant

species succession from diatoms to non-diatoms had also been found in the 1980s and

1990s (Lin et al., 2005). The water and sediment regulation of the Yellow River since

2002 may enhance primary production by increasing export of water and sediment to

the Bohai Sea. Changes of nutrient inputs from rivers in the semi-enclosed Bohai Sea

have larger and more long-lasting influence on the ecosystem than in other open seas

because the water residence time in the Bohai Sea is about 3 years (Liu et al., 2012),

There may be a close connection between the changes in nutrient loading and primary

production in coastal areas (Bernard et al., 2011). Recent studies also pointed out the

sensitivity of shelf seas to changing riverine loading due to anthropogenic

perturbations, but unfortunately these studies did not cover riverine Si input to coastal

marine ecosystems and consequences for the C cycle (Li et al., 2014; Woodland et al.,

2015). Our understanding of the regional coupled Si-C cycle and ecological effects of

changing river loadings in the continental shelves of eastern China is poor

(Ragueneau et al., 2010). In this paper we establish a Si and C budget for the Bohai

Sea to analyze the coupled Si-C biogeochemistry; the aim is to quantify the influence

of changing terrestrial loadings on the Si cycle and primary production in the Bohai

Sea.

**2. Materials and methods**

**2.1 Sampling and analytical methods**

Two campaigns were carried out in spring (May 3 to 24) and autumn (November 2 to

20) of 2012 at several sampling stations in the Bohai Sea and the adjacent area of the

Northern Yellow Sea (Fig. 1). Water samples in surface (0.5 m) and bottom water (<

2m from the sea floor) were collected using an oceanography water sampler (Seabird

911 CTD Plus, Sea-Bird Electronics, Bellevue, WA, USA). Ancillary parameters such

as temperature and salinity were recorded on board simultaneously. Also, surface (0-1





cm) sediment and core sediment samples (between 20 and 50 cm long) were collected

(Fig. 1b and 1c) in part of the stations.

The water samples were filtered with 200 μm Nylon sieves to remove zooplankton, subsequently filtered with 0.45 μm polyethersulfone filters, which were treated according to the following four steps: 1) cleaned with 1:1000 HCl for 24 h; 2) rinsed

with Milli-Q water to achieve a neutral pH; 3) oven-dried at 45°C for 72h; 4) weighed after cooling in a dryer with desiccant. The filters were stored at -20°C for determination of suspended particulate matter (SPM) and BSi, and filtrates were stored at 4°C after adding drops of chloroform for determination of DSi.

In addition, the pre-weighed water samples were filtered with 0.70 μm GF/F glass-fiber filters (Whatman, Maidstone, UK), which were also pre-cleaned according to the following four steps: 1) cleaned with 1:1000 HCl for 24 h; 2) rinsed with Milli-Q water to achieve a neutral pH; 3) burned at 450°C for 4h; 4) weighed after cooling in a dryer with desiccant. The filters were stored at -20°C for determination of

suspended particulate organic carbon (POC), and filtrates were stored at -20°C for determination of dissolved organic carbon (DOC) in a later stage.

Surface sediment samples (0-1 cm) were collected with a box sediment sampler after removing the overlying water, and then packed into sealed bags and frozen at -20 °C

for determination of BSi and total organic carbon (TOC). At the same time, sediment core samples were collected using a sampling tube with an inner diameter of 9 cm at some stations (Fig. 1b). Cores were divided into 1 cm intervals after overlying water was collected using syringes (13 mm 0.45 μm, PTFE) with needle tubing. The pore water of each subsample was separated by centrifugation and preserved as above for

DSi analyses; finally, subsamples were stored at -20°C before BSi and TOC analysis in a later stage.

Sampling expeditions were also carried out at the Lijin Station (Shandong Province)



at the Yellow River (Fig. 1a) during a full hydrological year in 2013-2014. Water

samples were collected for DSi, BSi, DOC and POC measurements once per month at

20 cm below the surface with at least 3 sampling points across the river main channel.

Water samples were pretreated as described above.

DSi was analyzed with a QuAAtro Autoanalyzer, using the silicomolybdic blue

method, with a detection limit of 0.030 $\mu$mol $l^{-1}$ and a relative standard deviation

<0.3%. BSi in SPM was extracted by NaOH solution (0.2 mol $l^{-1}$, 100 °C, 40 min ) and

corrected for mineral interferences using the Si:Al ratios (Ragueneau et al., 2005),

while the BSi content in sediment was measured using the alkaline extraction method

(1% $Na_2CO_3$, 85 °C, extraction during 8 hours, during which the extract is sampled

and analyzed every hour ) (DeMaster, 1981), with a measurement uncertainty of

0.25% and relative standard deviation <0.3%. Reactive silica (RSi) is the sum of DSi

and BSi.

DOC was determined using a high-temperature catalytic oxidation technique (Zhang

et al., 2013) with a TOC analyzer (TOC-$C_{CPH}$, Shimadzu, Japan); the relative standard

deviation is <2%. For POC determination, 3-5 drops of 2 mol $l^{-1}$ HCl were added to

the sample filters in a closed container with HCl fumes for 24 h to remove inorganic

carbon, and then dried at 45°C (Zhang et al., 2013). Subsequently, POC was

determined with an elemental analyzer (Euro Vector EA3000, Via Tortona, Milan,

Italy) with standard deviation <10%.

TOC in sediments was analyzed with the same elemental analyzer. Before

measurement, freeze-dried sediment samples were decalcified using 4 mol $l^{-1}$ HCl and

subsequently rinsed with de-ionized water (6–8 times) to achieve a neutral pH, and

then pretreated sediments were dried overnight at 60 °C (Hu et al., 2009) for TOC

determination. Replicate analyses of one TOC sample (n=6) provided a precision of

±0.02 %.



The software of Surfer 11.0 (Golden Software, inc. USA) and Origin 8.5 (OriginLab Corporation, USA) were used for mapping the concentration patterns in the Bohai Sea.

## 2.2 Water budget

The water budget of the Bohai Sea (Fig. 2) provides the basis for the calculation of the Si and C budgets. The hydrography of the Bohai Sea is largely determined by the Bohai Sea Coastal Current (BSCC) and exchange with the Yellow Sea. River discharge, precipitation, submarine groundwater discharge, surface runoff and evaporation are taken into account in the water budget calculation for the shelf in steady state as follows:

$$Q_R + Q_A + Q_{YTB} + Q_{GW} + Q_{SR} = Q_{BTY} + Q_{EVA} \qquad (1)$$

Where Q are water fluxes (km$^3$ yr$^{-1}$), subscripts R, A, YTB, GW, SR, BTY and EVA denote the river discharge, atmospheric deposition, Yellow Sea inflow, submarine groundwater, surface runoff , Bohai Sea outflow and evaporation, respectively (Table 1, Fig. 2). The estimated river input is about 34 km$^3$ yr$^{-1}$ for the 6 major rivers

discharging into the Bohai Sea (Table 1), and precipitation and evaporation amount to 34 km$^3$ yr$^{-1}$ and 85 km$^3$ yr$^{-1}$, respectively, based on Martin et al. (1993) and Lin et al. (2001). The water flux from the Bohai to Yellow Sea (BTY) is 470 km$^3$ yr$^{-1}$, and with a reverse flux (YTB) of 442 km$^3$ yr$^{-1}$ there is a net export 28 km$^3$ yr$^{-1}$ from the Bohai Sea to the Yellow Sea (Liu et al., 2003a). The submarine groundwater input is about

44 km$^3$ yr$^{-1}$ based on estimates of submarine groundwater discharge in the Yellow River delta (Peterson et al., 2008). The budget yields an estimate for surface runoff ($Q_{SR}$) of 1 km$^3$ yr$^{-1}$, which includes the discharge by small streams not included in the above river discharge.

## 2.3 Budget of reactive silica and organic carbon

The Si and C budgets of the Bohai Sea are estimated using a steady-state box model, focusing on the reactive Si (RSi, the sum of DSi and BSi) and OC in the water column



and accounting for the major hydrological, chemical and biological processes. In this

calculation, we use estimates for the fluxes of RSi and OC into and out of the Bohai

Sea, i.e. exchange through the Bohai Strait ($F_E$; $F_E$ = Input to the Bohai Sea ($F_{YTE}$) -

Output to the Yellow Sea ($F_{BTY}$)), riverine input ($F_R$), surface runoff ($F_{SR}$) from

surficial runoff and small rivers not included in $F_R$, submarine groundwater discharge

($F_{GW}$), atmospheric input ($F_A$), flux from porewaters ($F_B$) and sedimentation ($F_S$)

(Table1).


Internal processes such as primary production ($F_P$), regeneration ($F_{RC}$), respiration and

degradation are also taken into account. $F_E$, $F_B$, and $F_S$ are based on measurements

described in this paper, as well as the major contributions of Yellow River to $F_R$, while

the other fluxes are based on literature values. The various budget terms are discussed

in more detail below.

### 2.3.1 Riverine input

We estimated the DSi, BSi, DOC and POC fluxes for 6 major rivers discharging into

the Bohai Sea (Table 1 and Fig.1). Fluxes are calculated with long-time monitoring

data. On the basis of monthly data for BSi, POC and SPM, we found that the fraction

of BSi and POC in the suspended solids increases exponentially and linearly with

SPM concentration, respectively. The BSi and POC concentrations for rivers with

missing or scant data were estimated with the following regression equations:

$$C_{BSi} = a \times C_{TSS}{}^{b} \qquad (r=0.823, p<0.001) \qquad (2)$$

$$C_{POC} = c \times C_{TSS} + d \qquad (r=0.984, p<0.001) \qquad (3)$$

Where $C_{BSi}$ and $C_{POC}$ represent the BSi and POC concentration in the river ($\mu$mol l$^{-1}$),

respectively; $C_{TSS}$ represents the sediment content in the river (mg l$^{-1}$); $a$, $b$, $c$ and $d$

are constants, $a$ ($\mu$mol mg$^{-1}$) = 0.22435; $b$ (unitless) = 0.69235; $c$ ($\mu$mol mg$^{-1}$) =

0.0016; $d$ ($\mu$mol l$^{-1}$) = 2.9173. The average ratio DOC: POC in the Yellow River was

0.08, so DOC in other rivers without available data can be estimated from the POC

concentration.





### 2.3.2 Atmospheric deposition

Atmospheric input to the Bohai Sea was calculated from the DSi concentration in

precipitation (Martin et al., 1993; Zhang et al., 2004), dry deposition (Zhang et al.,

2004), combined with the area of the Bohai Sea (77300 km²). The POC in the air

mainly occurs in the particulate matter with grain size <2.5μm (Chen et al., 1997) and

the deposition rate of aerosol is about 0.001 m s⁻¹ (Duce et al., 1991); the POC

concentration in aerosol in the Bohai Sea was from the base station of Chang island in

the Bohai Sea Strait (Feng et al., 2007), and DOC in the coastal rainwater is from

Willey et al. (2000). Rainfall and aerosols have low BSi concentrations and can be

neglected as sources (Tréguer and De La Rocha, 2013).

### 2.3.3 Exchanges between the Bohai and Yellow Seas

Water exchange between the Bohai and Yellow Sea is driven by the BSCC in the

southwest of Bohai Sea and Yellow Sea Warm Current (YSWC) in the Northern

Yellow Sea (Fig. 1, Table 1). The RSi and OC fluxes through the Bohai Strait were

calculated using the water flux together with the measured RSi and OC concentration

data from the Southern Bohai Sea and the Northern Yellow Sea (Table 1).


### 2.3.4 Benthic flux at the sediment-water interface

The benthic flux of DSi at the sediment-water interface was calculated based on

Fick's first law (Berner, 1980) according to:

$$J_F = -\varphi \times D_s \times (\partial C / \partial C) \tag{4}$$

$$D_S = D_0 \times \varphi^{(m-1)} \tag{5}$$

Where $J_F$ represents the diffusion rate (mmol m⁻² d⁻¹); $\varphi$ is the porosity of the sediment

(dimensionless, 0.72–0.85, based on Liu et al. (2003b)); $D_s$ is the diffusion coefficient

in sediment (m⁻² d⁻¹); $C$ is the concentration (mmol l⁻¹), $z$ is the depth (m), $\partial C/\partial z$ is the

concentration gradient of DSi at the sediment-water interface; $D_0$ is the molecular

diffusion coefficient of solute in infinitely diluted solutions (m⁻² d⁻¹, Li and Gregory



(1974)); $m$ is an empirical coefficient (dimensionless, for $\varphi \leq 0.7$, $m = 2$; for $\varphi > 0.7$, $m = 2.5$–$3.0$) (Ullman and Aller 1982).

As no direct measurement data are available, we estimated the DOC flux from the
benthic flux of DSi to the water column based on the molar ratio of BSi : TOC in surface sediments of 0.56.

### 2.3.5 Sedimentation

Sedimentation of BSi and OC in the Bohai Sea were calculated from the accumulation
rate and the surface area (Ingall and Jahnke, 1994; Liu et al., 2005). The accumulation rates were based on the following equations:

$$R_{BSi} = C_{BSi} \times MAR / 28 \tag{6}$$

$$R_{OC} = C_{TOC} \times MAR / 12 \tag{7}$$

where $R_{BSi}$ and $R_{OC}$ represent for the accumulation rates of BSi and OC (mol m$^{-2}$ yr$^{-1}$);
$C_{BSi}$ and $C_{TOC}$ represent for the BSi and TOC content in surface sediments (%); $MAR$ is the mass accumulation rate of the sediment (g m$^{-2}$ yr$^{-1}$); 28 and 12 are the molar weight of Si and C, respectively.

### 2.3.6 Submarine groundwater discharge and surface runoff

The submarine groundwater DSi flux into the Bohai Sea was calculated from the water flux obtained from $^{228}$Ra and $^{226}$Ra mass balance models (Peterson et al., 2008) and the DSi concentration in groundwater (Lin et al., 2011). As there are no data on DOC input to the Bohai Sea via submarine groundwater, we assumed that the DOC concentration in submarine groundwater equals to that in rivers based on Barrón et al.
270    (2015).

Similar to the water budget, DSi and OC input from surface runoff and rivers not included in the large river inputs (Table 1) were obtained as a result of the budget calculation.






### 2.3.7 Primary production

Primary production was estimated from the average primary production in the euphotic layer, obtained by integrating the seasonal data from 1998 to 2008 estimated for the total area of the Bohai Sea by satellite remote sensing technology calibrated

against measured productivity (Tan et al., 2011). The rates of DSi uptake by phytoplankton and BSi regeneration rate were calculated using the Redfield ratio (C:Si=106:15, atom basis, Brzezinski, 1985); OC respiration was calculated according to Wei et al. (2004), who demonstrated that respiration accounted for 78% of primary production in the Bohai Sea.


## 3. Results

### 3.1 Distribution of RSi and OC in the water column

The DSi concentrations strongly vary in space and time; those in fall exceed those in

spring (Table 2). In spring, the distribution of DSi in surface water is similar to that in bottom water, and DSi concentrations are fairly low in the Northwestern part of the Bohai Sea, but high in the southeastern part, particularly in the Bohai Strait. In autumn, DSi concentration in surface water is lower in the central part of Bohai Sea than in other areas, with fairly high levels in the Laizhou Bay and Bohai Strait (Fig.

295     3).

The BSi concentrations are similar to those of DSi; in fall the BSi concentration exceeds that in spring by a factor of four (Table 2). In spring, surface water BSi concentrations are fairly high in the Bohai Bay, Laizhou Bay and Yellow River estuary,

and lower in the central part of the Bohai Sea. The distribution of BSi in the bottom water differs from that in the surface water, with relatively high BSi concentrations occurring in the central area of the Bohai Sea. In autumn, the distribution of BSi in surface water is similar to that in bottom water, with fairly high concentrations in the Yellow River estuary and other nearshore areas (Fig. 3).


The DOC concentration in fall is slightly higher than that in spring (Table 2). In

spring, the DOC concentration in surface water is lower than in bottom water, but the

spatial distributions of DOC in autumn and spring are similar, with fairly high

concentrations in the nearshore and low ones in the offshore areas. In fall, DOC

concentrations show only small spatial variability (Fig. 3).

The POC concentrations and their spatial distributions in spring are close to that in

fall (Table 2), with fairly high concentration in the western part of the Bohai Sea and

the Yellow River estuary (Fig. 3).


**3.2 Distribution of RSi and OC in the sediment**

The BSi content in the surface sediments is 0.4% and the TOC content in the surface

sediments is 0.3% with a large variability in both BSi and TOC (Table 3). The spatial

pattern and variability of BSi in sediments is similar to that of TOC, with high

concentrations in the mud area of the Bohai Sea and the area adjacent to the Yellow

River estuary (Fig. 4).

**3.3. Budget of RSi and OC in the Bohai Sea**

The estimated riverine RSi and OC fluxes are, respectively, 5.0 Gmol yr$^{-1}$ and 38

Gmol yr$^{-1}$, and DSi and DOC account for 54% and 9% of total RSi and OC fluxes,

respectively.

The estimated deposition flux in the Bohai Sea is 0.2 Gmol yr$^{-1}$, primarily (90%) from

wet deposition. OC from atmospheric deposition is 4.2 Gmol yr$^{-1}$, with an important

contribution (60%) from wet deposition.

The inputs of RSi and OC from the Yellow Sea into the Bohai Sea are 3.2 Gmol yr$^{-1}$

and 120 Gmol yr$^{-1}$, respectively, while the output RSi (3.5 Gmol yr$^{-1}$) and OC (140

Gmol yr$^{-1}$) fluxes from the Bohai Sea to the Yellow Sea are similar resulting in net





outputs of RSi and OC from Bohai Sea of 0.3 Gmol yr$^{-1}$ and 20 Gmol yr$^{-1}$, respectively.

Internal cycling of Si and OC are important terms in the budget. Based on primary production in the euphotic zone of the Bohai Sea, C sequestration is 3280 Gmol yr$^{-1}$,

which means that 460 Gmol yr$^{-1}$ of BSi is ingested according to the Redfield ratio, and 2560 Gmol of OC (78%, see 2.3.7) is consumed by respiration. The estimated sedimentation fluxes of BSi and OC are 30 and 60 Gmol yr$^{-1}$, respectively. Recycling of BSi in the water column amounts to 430 Gmol yr$^{-1}$ of DSi released. Biodegradation and photo oxidation of OC in the water column is about 3220 Gmol yr$^{-1}$.


The benthic fluxes of DSi and DOC at the sediment-water interface are further important sources of respectively 15 Gmol yr$^{-1}$ and 27 Gmol yr$^{-1}$. The calculated submarine groundwater discharge into the Bohai Sea (Table 1) amounts to 8.0 Gmol yr$^{-1}$ for DSi and 4.4 Gmol yr$^{-1}$ for DOC. The estimated surface runoff fluxes are 2.1

Gmol yr$^{-1}$ for RSi and 6.4 Gmol yr$^{-1}$ for OC.

## 4. Discussion

### 4.1 Distribution of RSi and OC in the water column

The DSi distribution is largely affected by the circulation system of the Bohai Sea, particularly in the area adjacent to the Bohai Strait. The high DSi concentration in the southeastern part of the Bohai Sea is due to high DSi in the water mass coming in from the Northern Yellow Sea through the Bohai Strait. The DSi distribution is also influenced by the terrestrial input, particularly in the area near the mouth of the

Yellow River (Fig. 3).

The BSi concentration in the Bohai Sea is similar to that in the other parts of the Eastern China Sea (Liu et al., 2005). BSi is an important component (30%) of RSi in the Bohai Sea, which is lower than in the Yellow River water (52%, Ran et al., 2015)



but higher than that in the Changjiang Estuary (8%; Gao et al., 2013).The rivers

draining into the Bohai Sea carry abundant BSi and have a large influence on the

composition of RSi. In addition, the distribution and transportation of BSi are affected

by sediment resuspension (Liu et al., 2005), which may be the main reason why BSi

in the bottom water exceeds that in the surface water in parts of the Bohai Sea.


DOC is the dominant (>95%) form of OC in the world's oceans (Reeburgh et al.,

1997), and a little less so (89%) in the Bohai Sea. The spatial distributions of DOC

and POC are similar in the Bohai Sea, both being affected by the same processes, such

as input from land by rivers, primary production, biological action, sediment

resuspension and many other factors. Our results show a significant negative

correlation between POC concentrations and salinity ($r = -0.430$, $p < 0.05$, in spring; $r = -0.348$, $p < 0.01$, in autumn), indicating that POC concentrations in the coastal areas

exceed those in the high salinity waters and that the distribution of POC is largely

determined by the terrestrial input. The POC distribution is also affected by sediment

resuspension in ocean margins (Zhu et al., 2006), which may explain why POC in

bottom water is generally higher than in surface water.

The average molar Si : C ratio of BSi and POC in the Bohai Sea of 0.12 is close to

that of diatoms in coastal waters (Brzezinski, 1985). This means that BSi is mainly

from marine primary production by diatoms, which is consistent with the results from

Jiaozhou Bay (Liu et al., 2008a) and East China Sea (Liu et al., 2005). The C : N

atomic ratio in SPM ranges from 1 to 10, with an average value of 5, indicating that

OC also originates from marine phytoplankton production.

**4.2 Distribution of RSi and OC in sediments**

There are no differences of both BSi and TOC among seasons at the 95% confidence

level. The BSi content of surface sediments in the Bohai Sea is similar to that in the

continental shelves in Eastern China (Liu et al., 2009), but lower than in the



 northwestern Indian Ocean (Koning et al., 1997), Southern Ocean (Van Cappellen and

Qiu, 1997) and the equatorial Pacific Ocean (Piela et al., 2012).

High concentrations of both BSi and TOC concentrations in the mud area of the Bohai

Sea (Fig. 4) suggest that the sediment grain size and hydrodynamic setting have an

important influence on the preservation of BSi. BSi content in the sediment is much

lower than that in SPM (0.1%-3.0%, average 0.8%), which indicates that BSi in

particles has been degraded during sedimentation and burial. Meanwhile, the average

Si : C ratio in sediments of 0.56 is much higher than that in suspended particulate

matter. This confirms that degradation rate of OC in the ocean is faster than that of

BSi (Ragueneau et al., 2000) due to the lower preservation efficiency of autogenetic

OC than that of autogenic BSi (Muller-Karger et al., 2005; Tréguer and De La Rocha,

2013).

### 4.3 Budget of RSi and OC in the Bohai Sea


The budget of RSi shows that the benthic flux across the sediment-water interface is

the major source of reactive Si in the Bohai Sea water mass, contributing 49% of the

total RSi input (Table 1, Fig. 5). The next largest source is from submarine

groundwater, comprising 26% of total inputs. The river input accounts for 17%, and

all other inputs are minor (surface runoff, 7%; atmospheric deposition, <1%). The

dominant output fluxes of RSi in the water column is by the sedimentation and export

to the Yellow Sea, contributing 99% and 1% to Si removal in the budget, respectively.

Overall, considering all exogenous input of OC into the Bohai Sea, riverine flux alone

accounts for 47%, followed by the benthic flux of DOC, accounting for 34%; the

remaining 19% is from surface runoff (8%), submarine groundwater discharge (6%),

and atmospheric deposition (5%). The dominant outputs of OC in the Bohai Sea are

sedimentation (75% of total output) and the outflow to the Yellow Sea (25%).





The BSi share in river export in total RSi of 46% is much higher than the average

value for global rivers (15%) (Laruelle et al., 2009). POC comprises 90 % of the

riverine OC, which also exceeds the average for world rivers of 40% (Hedges et al.,

1997). The Yellow River export to the Bohai Sea is 68% of total exogenous input for

RSi and 75% for OC.


The water exchange between the Bohai and Yellow Seas has only a minor influence

on the budget of RSi and OC; however, it has an important effect on the distribution,

transport, transformation and retention time of RSi and OC.

The benthic recycling of Si in the sediment is a particularly important flux into the

DSi pool in the water column, which confirms earlier studies (Van Cappellen et al.,

1997). DSi concentration gradients in the pore water at all studied stations show a

diffusion flux from sediment to water column. The diffusion rates vary from 0.38 to

0.62 mmol $m^{-2}$ $d^{-1}$, similar to previously reported data (Liu et al., 2011). The high

benthic flux plays an important role in maintaining the level of primary production in

the water column and also results in a concentration gradient, with higher DSi

concentration in bottom than in surface waters.

The DOC in pore water is also an important source of DOC in the water column

(Burdige et al., 1999; Barrón et al., 2015). Another way to estimate the benthic DOC

flux is by assuming DOC diffusion rates to be similar to those in bare sediments (0.9

mmol $m^{-2}$ $d^{-1}$) (Burdige et al., 1999). This yields a DOC flux of 26 Gmol $yr^{-1}$, which

confirms our estimate (27 Gmol $yr^{-1}$) based on the assumed BSi : TOC ratio of 0.56.

According to the difference of the diffusion flux of DSi and BSi sedimentation, the

net burial flux of BSi is 15 Gmol $yr^{-1}$, which is 3.3% of the total primary production.

This large BSi sedimentation flux exceeds the average value for the world ocean

(2.6%) (Tréguer and De La Rocha, 2013). The gross burial efficiency of BSi is 50%

in the Bohai Sea, similar to the East China Sea of 36−97% (Liu et al., 2005) and





higher than the average of 17−20 % in the world's oceans (Bernard et al., 2010;

Tréguer and De La Rocha, 2013).

The net burial of OC is 33 Gmol yr$^{-1}$ or 1.0% of the primary production, which is also

much higher than the world ocean (0.3%, Muller-Karger et al., 2005). This shows that

the Bohai Sea is a potential sink for both Si and C.

Previous studies showed that nutrient inputs from the submarine groundwater were

1–2 times higher than those associated with river discharge into the Bohai Sea (Liu et

al., 2011). Relative contributions from submarine groundwater and riverine Si input

are similar to those in the Yellow Sea (Kim et al., 2005) and Mediterranean Sea

(Rodellas et al., 2015), a semi-closed sea like the Bohai Sea. Our estimate for

submarine groundwater DSi input exceeds riverine input, and this agrees with the

above studies, while the DOC input from submarine groundwater is less important

than riverine input.


**4.4 Response of primary production to changing riverine RSi transport**

The DSi concentration in the Bohai Sea had decreased in the period 1980–1990, and

has been stable more recently. The average DSi concentration in the Bohai Sea in the

early 2000s was only 1/3 of that in 1980s (Tang et al., 2003; Ning et al., 2010; Liu et

al., 2011; Liu, 2015). Meanwhile, the DIN concentration increased from 1.7 μmol l$^{-1}$

in the 1980s (Tang et al., 2003) to 5.1 μmol l$^{-1}$ in 2000 (Li et al., 2003) and 10.6 μmol

l$^{-1}$ in 2012. Nutrient stoichiometry has changed significantly with molar Si : N ratios

varying from 14 in 1980s to 1.5 in 2000 and 0.6 in 2012, respectively.

The Bohai Sea has therefore changed from an N limited ecosystem in the 1980s to a

Si limited system in recent years, and the BSi production by diatoms now largely

depends on available silica (Tang et al., 2003; Ning et al., 2010; Liu et al, 2011). The

Yellow River discharge represents more than 70% of the total freshwater discharge

into the Bohai Sea (33 km$^3$ yr$^{-1}$, Table 1). Since the water residence time in the Bohai





Sea is about 3 years (Liu et al., 2012), changes of riverine Si input in the Bohai Sea

would have long-lasting influence on the ecosystem's functioning. Statistical analysis

suggests that there is a significant relationship between DSi and RSi flux of the

Yellow River in year n-1 and primary production in the Bohai Sea in year n (DSi:

$p<0.005$; RSi: $p=0.02$) (Fig.6 and 7) reflecting the long residence time of Si. This also

suggests that changing terrestrial Si loadings have a direct and long-time influence on

primary production in the Bohai Sea.

Since 2002, the water discharge and sediment load of the Yellow River have increased

significantly compared with the late 1990s due to the water and sediment regulation

(Fig. 6). The annual DSi flux of Yellow River in July increased 5–10 fold and RSi flux

by a factor of 3 since 2002; since this year sediment load and water discharge

regulation in spring has led to peak events (Fig. 6) (Gong et al., 2015; Liu, 2015).

Using a factor of 3 increase of RSi river export (from 0.9 prior to 2002 to 4.3 G mol

$yr^{-1}$ at present, see Fig. 5) with the regression equation in Fig. 7 results in an increase

of primary production by 10% since 2002 in comparison with the levels in 2000 and

2001. This is confirmed by data indicating that , DOC in the Bohai Sea increased from

2.1 mg $l^{-1}$ before (Zhang et al., 2006) to 2.6 mg $l^{-1}$ (Chen, 2013) and 3.9 mg $l^{-1}$ (this

study) after the Yellow River water-sediment regulation in spring., the TOC

concentrations in the Bohai Sea have been increasing in the same period, which

indicates that increasing Si loadings may enhance both TOC and DOC levels in the

Bohai Sea, particularly in the part close to the river mouth.

## 5. Conclusions

The distributions of RSi and OC in the Bohai Sea show seasonal and regional

variation, and are mainly affected by the riverine input, primary production and water

exchange between the Bohai Sea and Yellow Sea. BSi and TOC are mainly from

marine primary production, and areas with high BSi and TOC contents in the

sediments are mainly in the estuarine and mud areas.



The benthic diffusion in the Bohai Sea is the major source of external Si to the water
column, accounting for 49% of the exogenous Si inputs, followed by the submarine
groundwater discharge (26%), riverine input (17%), surface runoff (7%), and
atmospheric deposition (<1%). The dominant removal processes of RSi in the Bohai
Sea are BSi sedimentation (99% of total output) and the outflow of RSi to the Yellow
Sea (1%).

The riverine flux contributes 47% of all exogenous OC input to the Bohai Sea,
followed by benthic flux of DOC, accounting for 34%; surface runoff (8%),
submarine groundwater input (6%), and atmospheric deposition (5%) represents the
remaining 19%. The dominant outputs of OC in the Bohai Sea are sedimentation
(75% of total output) and the outflow to the Yellow Sea (25%). The Bohai Sea is a
sink for both Si and C, net burial of BSi and OC in sediments amounting to 3.3% and
1.0% of primary production, respectively.

DSi in the Bohai Sea had decreased and then maintained stable in the last three
decades. Earth surface process modified by human activities and riverine load
variations change the exogenous Si input and thus primary production. Primary
production in the Bohai Sea has increased by 10% since 2002, as a result of the
increasing riverine RSi input from the Yellow River due to water-sediment regulation.


A quantitative mechanistic understanding of the key processes controlling Si flow and
preservation of C in the land–ocean continuum is needed. The mechanistic
understanding is necessary to parameterize the various processes involving C and Si
and their sensitivity to external perturbations at the larger scales of earth system
models. At present, this lack of understanding limits our ability to predict the present
and future contribution of the aquatic continuum fluxes to the global C and Si budget.





**Acknowledgements**

This study was supported in part by the Natural Science Foundation of China
(Project No. 41106072 and 41376093). We would like to thank Tao Sun, Hong Che
and Lili Zheng for their assistance in the field sampling. We are grateful to Prof.
Ruixiang Li for providing us with data on chlorophyll and primary production. Data
from the cruises presented in this paper can be obtained from the corresponding
author.





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



**Table 1.** Main fluxes of reactive silica and organic carbon budget in the Bohai Sea.

| Flux | Parameters to estimate Si-C fluxes | | | | | | | |
|---|---|---|---|---|---|---|---|---|
| Riverine input | Water discharge | Sediment content | DSi | BSi | DOC | POC | RSi flux | OC flux |
| ($F_R$) | km³ yr⁻¹ | mg L⁻¹ | µmol L⁻¹ | µmol L⁻¹ | µmol L⁻¹ | µmol L⁻¹ | Gmol yr⁻¹ | Gmol yr⁻¹ |
| Yellow River | 22.4 [1] | 6868 [1] | 81.4±24.7 [2] | 70.0±16.3 [2] | 151–280 [3] | 235–6980 [4] | 3.39±0.90 | 28.5±9.8 [4] |
| Daliao River | 4.31 [1] | 1592 [1] | 52.7 [5] | 37.0 | 36.4 | 455 | 0.39 | 1.97 |
| Shuangtaizi River | 3.95 [5] | 5063 [5] | 105 [5] | 82.3 | 73.4 | 918 | 0.74 | 3.92 |
| Hai River | 0.90 [1] | 88.9 [1] | 29.3 [6] | 5.0 | 20.4 | 255 | 0.03 | 0.25 |
| Luan River | 0.60 [7] | 100 [7] | 87.2 [5] | 5.4 | 20.5 | 256 | 0.06 | 0.17 |
| Daling River | 2.06 [8] | 8592 [8] | 77 [5] | 119 | 111 | 1389 | 0.40 | 3.09 |
| Atmospheric deposition | Precipitation | Si concentration in the rainfall | Dry deposition rate of Si | DOC concentration in the rainfall | Dry deposition rate of POC | RSi flux | OC flux | |
| ($F_A$) | km³ yr⁻¹ | µmol L⁻¹ | µmol m⁻² yr⁻¹ | µmol L⁻¹ | mol m⁻² yr⁻¹ | Gmol yr⁻¹ | Gmol yr⁻¹ | |
| | 33.9 [9] | 4.1±0.9 [10] | 72.3 [10] | 75±34 [11] | 0.02±0.01 [12] | 0.2±0.1 | 4.2±2.2 | |
| Primary production | Primary productivity of carbon | Primary productivity of BSi | Si fixed by primary production | C fixed by primary production | | | | |
| ($F_P$) | g m⁻² yr⁻¹ | mol m⁻² yr⁻¹ | Gmol yr⁻¹ | Gmol yr⁻¹ | | | | |
| | 509±38 [13] | 6.0±4.5 | 460±35 | 3280±250 | | | | |
| Water exchange | Water discharge | DSi concentration | BSi concentration | DOC concentration | POC concentration | RSi flux | OC flux | |
| ($F_E$) | km³ yr⁻¹ | µmol L⁻¹ | µmol L⁻¹ | µmol L⁻¹ | µmol L⁻¹ | Gmol yr⁻¹ | Gmol yr⁻¹ | |
| Output to YS ($F_{BTY}$) | 470 [14] | 5.3±3.9 [15] | 2.2±3.4 [15] | 271±80.8 [15] | 31.7±5.0 [15] | 3.5±3.4 | 140±40 | |
| Input to BS ($F_{YTE}$) | 442 [14] | 5.5±4.3 [16] | 1.8±2.2 [16] | 249±15.8 [16] | 22.5±10.0 [16] | 3.2±2.9 | 120±11 | |
| Flux from porewater | Diffusion rate of DSi | Diffusion rate of DOC | Benthic DSi flux | Benthic DOC flux | | | | |
| ($F_B$) | mmol m⁻² d⁻¹ | mmol m⁻² d⁻¹ | Gmol yr⁻¹ | Gmol yr⁻¹ | | | | |
| | 0.53±0.30 [17] | 0.95±0.54 [18] | 15±8.5 | 27±15 | | | | |
| Sedimentation | Accumulation rate | BSi content | TOC content | Accumulation rate of BSi | Accumulation rate of TOC | BSi flux | TOC flux | |
| ($F_S$) | g cm⁻² yr⁻¹ | % | % | mol m⁻² yr⁻¹ | mol m⁻² yr⁻¹ | Gmol yr⁻¹ | Gmol yr⁻¹ | |
| | 0.1–0.6 [19] | 0.2–0.7 | 0.1–0.7 | 0.2–0.8 | 0.2–1.8 | 30±12 | 60±13 | |
| Submarine groundwater discharge | Water discharge | DSi | DOC | DSi flux | DOC flux | | | |
| ($F_{GW}$) | km³ yr⁻¹ | mg L⁻¹ | mg L⁻¹ | Gmol yr⁻¹ | Gmol yr⁻¹ | | | |
| | 40.4–46.7 [20] | 183±41.6 [21] | 100±15 [22] | 8.0±1.8 | 4.4±0.7 | | | |

[1] Ministry of Water Resources of the People's Republic of China, 2013; [2] From this study of Kenli station in the Yellow River in 2013–2014; [3] Wang et al. (2012) and Zhang et al. (2013); [4]Wang et al.





(2012), Zhang et al. (2013) and this study; [5] Liu et al. (2009); [6] Liu et al. (2008b); [7] Li and Feng (2007); [8] Dou et al. (2014); [9] Lin et al. (2001); [10] Zhang et al. (2004); [11] Willey et al. (2000); [12] Duce et al. (1991), Chen et al. (1997) and Feng et al. (2007); [13] Tan et al. (2011); [14] Liu et al. (2003a); [15] Data for the southern Bohai Sea are from this study; [16] Data from the northern Yellow Sea of this study; [17] From Liu et al. (2011) and this study; [18] Calculated based on the Si/C in the surface sediment; [19] Hu et al (2016); [20] Peterson et al. (2008); [21] Chen et al. (2007); [22] Average value of DOC in the rivers flowing into the Bohai sea; The unidentified data are from this study and calculations; surface runoff (non-river compartment) ($F_{SR}$) and other internal processes are results of the budget calculation; Number following ± are standard deviations.





**Table 2.** Reactive silica and organic carbon concentrations in surface water (0.5 m), bottom water (< 2m from the sea floor) and average for water column in the Bohai Sea in 2012.

| Season | Layer | RSi [a] | | OC [a] | |
|---|---|---|---|---|---|
| | | DSi | BSi | DOC | POC |
| | | μmol l⁻¹ | | | |
| Spring | Surface | 4.0±4.2 | 0.7±1.0 | 265±100 | 35±14 |
| | Bottom | 3.6±3.7 | 1.5±1.5 | 352±193 | 35±13 |
| | Whole layer | 3.8±3.9 | 1.1±1.3 | 327±167 | 35±13 |
| Fall | Surface | 7.0±3.0 | 3.8±4.0 | 225±49 | 24±12 |
| | Bottom | 7.6±3.3 | 4.2±4.2 | 206±39 | 33±12 |
| | Whole layer | 7.3±3.1 | 4.0±4.0 | 217±47 | 28±19 |

[a] Mean ± standard deviation.





**Table 3.** Biogenic silica and total organic carbon contents of the surface sediment and core sediment in the Bohai Sea.

| Sample and station [a] | | BSi (%)[b] | | TOC (%)[b] | |
|---|---|---|---|---|---|
| | | Range | Average | Range | Average |
| Surface sediment | Spring | 0.29–0.61 | 0.41±0.12 | 0.10–0.66 | 0.31±0.20 |
| | Fall | 0.20–0.69 | 0.38±0.15 | 0.10–0.67 | 0.35±0.19 |
| Core sediment | B45 | 0.34–0.59 | 0.45±0.07 | 0.13–0.84 | 0.36±0.14 |
| | B49 | 0.20–0.46 | 0.28±0.05 | 0.10–0.62 | 0.23±0.13 |
| | B61 | 0.42–0.93 | 0.59±0.11 | 0.10–0.75 | 0.45±0.15 |

[a] See Fig. 1 for the location of the stations; [b] percentage of sediment by weight (%).



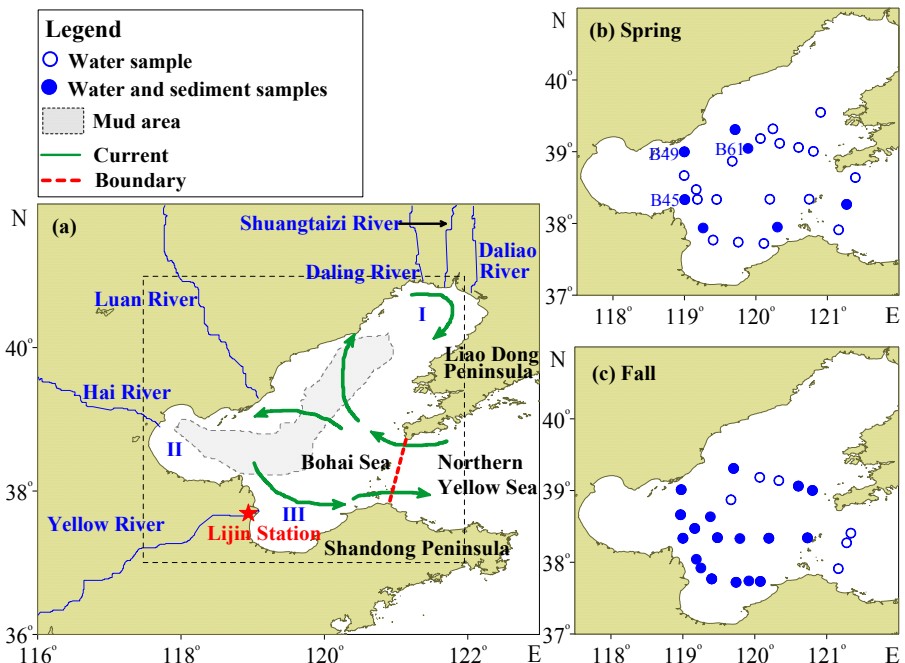

**Figure 1.** Rivers, mud area, circulation system and sampling stations in the Bohai Sea (I = Liaozhou Bay, II = Bohai Bay, and III = Laizhou Bay), and Mud area and circulation system are redrawn according to studies of Hu et al. (2012) and Sündermann and Feng (2004).




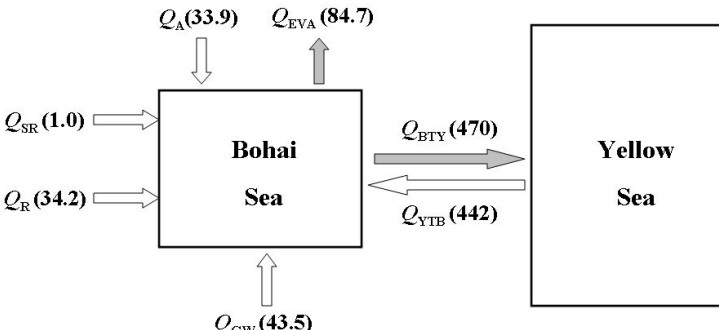

**Figure 2.** Water budget in the Bohai Sea. Fluxes in $km^3 \ yr^{-1}$. Subscripts R, A, YTB, GW, SR, BTY and EVA denote the river discharge, atmospheric deposition, Yellow Sea inflow, submarine groundwater, surface runoff , Bohai Sea outflow and evaporation, respectively.





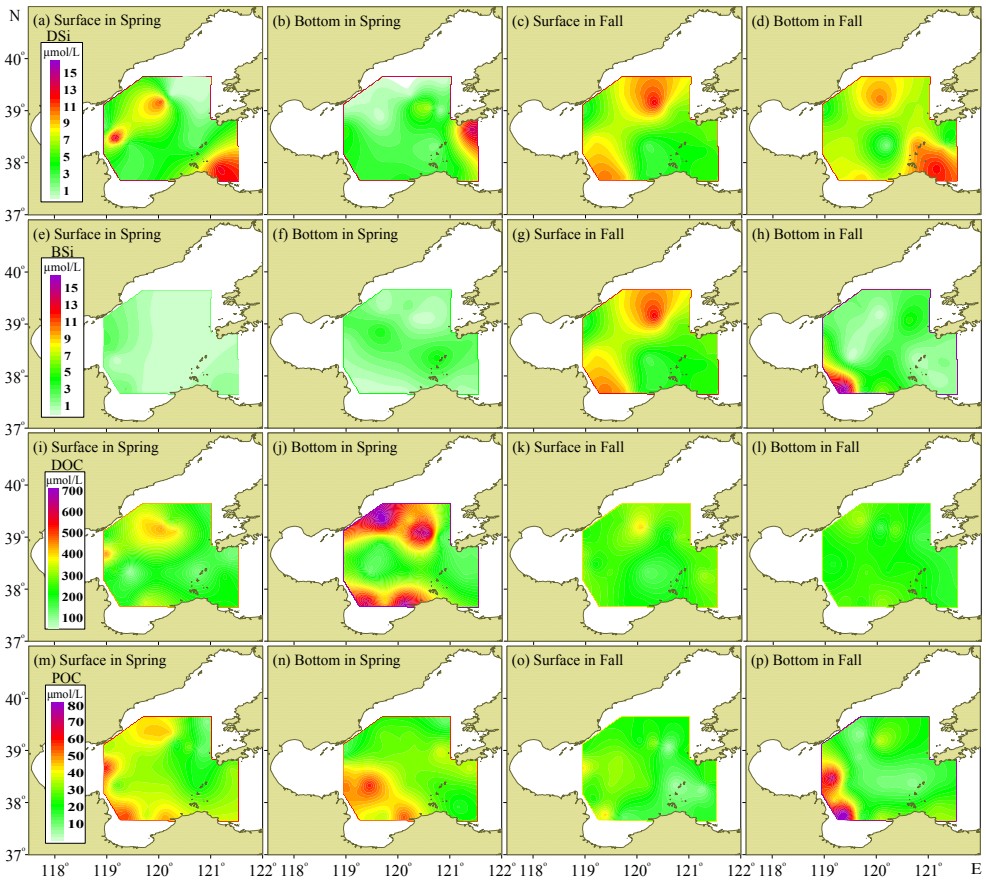

**Figure 3.** Spatial distributions of DSi (a-d), BSi (e-h), DOC (i-l) and POC (m-p) in the Bohai Sea in surface

and bottom water for spring and fall in 2012.

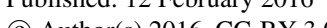



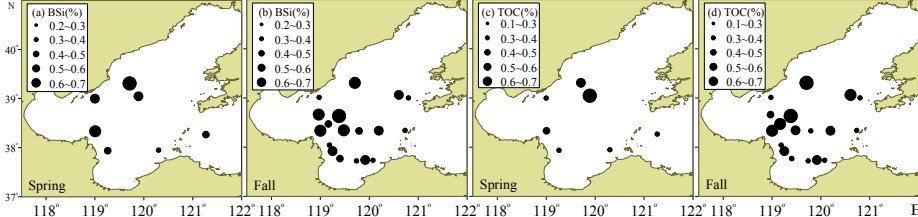

**Figure 4.** Distributions of total OC (a–b) and BSi (c–d) in the surface sediment of the Bohai Sea.





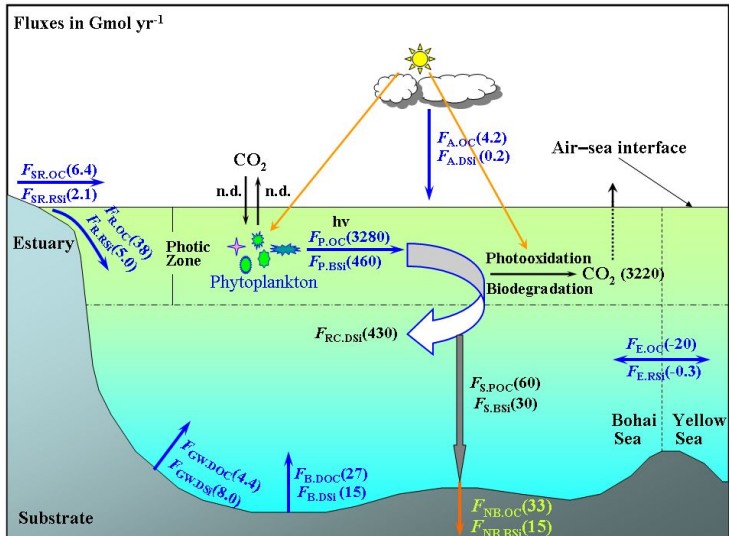

**Figure 5.** Fluxes of reactive silica (DSi and BSi) and organic carbon (DOC, POC) in the Bohai Sea. $F_A$: Atmospheric deposition; $F_B$ = Benthic diffusion flux; $F_R$: River input; $F_E$ = Water exchange from Yellow Sea to Bohai Sea ($F_E = F_{YTE} - F_{BTY}$; negative fluxes denote outflow from Bohai Sea to Yellow Sea); $F_{GW}$ = Submarine groundwater discharge; $F_{NB}$ = net burial; $F_P$ = Primary production; $F_{RC}$ = Internal recycle; $F_S$ = Sedimentation; $F_{SR}$ = Surface runoff (small rivers not included in $F_R$); n.d.: No data available.




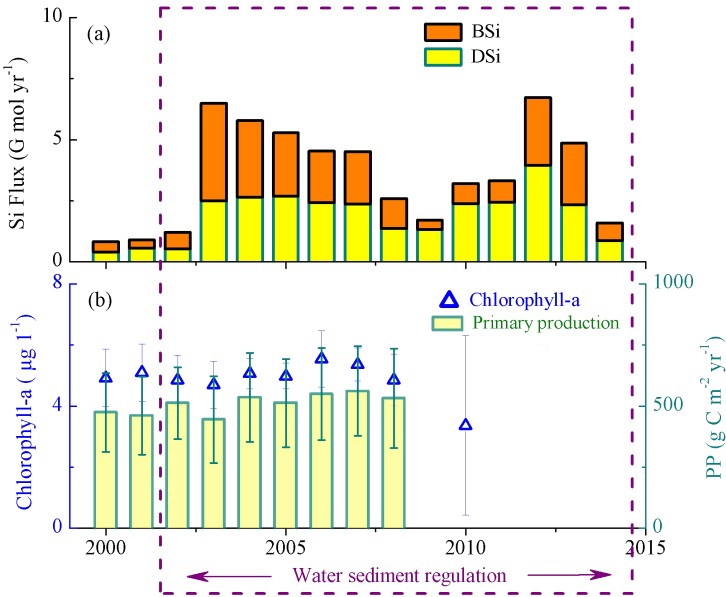

**Figure 6.** Data for Lijin station in the lower Yellow River for the period 2000-2015 representing

(a) DSi and BSi fluxes; (b) data for chlorophyll-*a* and primary production in the Bohai Sea with

standard deviation. DSi in the Yellow River is from Gong (2015), Ran et al. (2015) and this study.

BSi data is calculated by Equation 2. Chlorophyll-*a* and primary production (*PP*) in 2000-2008 of

the Bohai Sea are from Tan et al. (2011), data for 2010 are from Chen et al. (2013) and Zhao et al.

(2015).





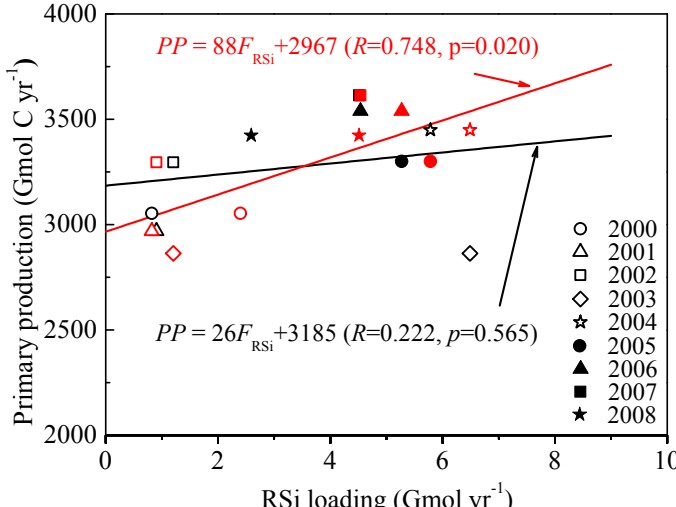

**Figure 7.** Relationship between annual RSi loading of the Yellow River and primary production in the Bohai Sea obtained by linear regression (black markers represent *PP* and silica loading in the same year, red markers represent *PP* in year n and RSi loading for year n-1. PP data correspond to the data in Figure 6 recalculated to Gmol C yr$^{-1}$. The regression equation is $PP=a\,F_{RSi}+PP_0$, with *PP* is the primary production (Gmol C yr$^{-1}$) and $PP_0$ is the intercept, representing the background primary production from all RSi sources except the Yellow river; $F_{RSi}$ is the RSi flux of the Yellow River (G mol yr$^{-1}$); *a* is a constant (Gmol Gmol$^{-1}$).