# Peer review of "The silica-carbon biogeochemical cycle in the Bohai Sea and its responses to the changing terrestrial loadings"

_Biogeosciences, 2016_

## Referee Comment (RC1) · Anonymous Referee #1 · 21 Mar 2016

**Review of manuscript bg-2016-42**

The manuscript by Jun Liu and co-workers presents results from field measurements of Si and C in the water column and sediment of Bohai Sea. They estimated the Si and C budgets and assessed the impact of change in terrestrial loadings on primary production in the Bohai Sea.

Quantifying Si and C budgets and fluxes in the Bohai Sea, which is important for understanding the anthropogenic impacts on coastal biogeochemical cycles and ecosystem function, is a significant contribution to the field. My main concern is that the description and explanation of the Si and OC distributions are not well structured and supported. In addition, they are old news already. Therefore, there is some room for improvement in the focus of the manuscript before it is suitable for publication.

*General comments*

**1. Manuscript focus**

The distribution of silicon along with other nutrients in the Bohai Sea has been systematically studied by Li et al. (2003), Zhang et al. (2004) and Liu et al. (2011) and so on. It has been demonstrated that the Si distribution is highly influenced by diffusion from sediment and major river input (Li et al., 2003; Zhang et al., 2004). In general, the Si concentration is higher in the coastal areas than in the central Bohai Sea and higher in near-bottom waters than in surface waters (Zhang et al., 2004; Liu et al., 2011; also shown in this study). Nutrients including Si are vertically well mixed in the coastal areas, while stratification takes place in the upper water of the central Bohai Sea especially in summer (Li et al., 2003; Liu et al., 2011). The Si concentration in the Bohai Sea is usually higher in autumn than in spring (Zhang et al., 2004; also shown in this study). Unfortunately, I couldn't recognize new added value in this manuscript to the current understanding of the nutrients distribution in the Bohai Sea. I would suggest that the authors rather focus on the nutrient budgets estimate and impact study, and eliminate the nutrients distribution part. Unless the authors make a clear point on what is novel in their findings.

**2. Structure**

The authors may want to consider reorganizing the "2.1 sampling and analytical methods" section by subsections (e.g. 2.1.1 sampling, 2.1.2 pre-treatment of samples and 2.1.3 laboratory analysis) or grouping the related paragraphs together. L128-131 should be merged into the first paragraph of section 2.1. This would help readers to find information more efficiently.

The same for Section 4.3 that some paragraphs can be grouped together, otherwise information gets fragmented.

*Specific comments:*

L25: They are old news already. See General comments 1.

L27: These are interesting results! State more numbers like 49% DSi input is from benthic diffusion and so on.

L40-41: Those estimates are likely upper limits of diatom contributions to the primary productivity, globally, as stated by Nelson et al. (1995). I would suggest to use "up to" instead of "more than".

L51-53: This statement cannot be supported by the two references listed here, especially the one by Conley (1997). Please provide appropriate references.

L56-58: Do you mean it will influence the function of coastal ecosystem as a biological pump?

L63: "have induced *significant* changes"

L69: "phytoplankton composition *in the Bohai Sea*"

L73: "... may *have enhanced* primary production by increasing export of  *nutrients* and sediment to the Bohai Sea."

L75: "than in  open seas"

Section 2.1: How many replicants of water samples and sediment samples did you measure? Please state that in this section.

L165-166: "The hydrography …" This sentence belongs to introduction.

L172: I would use "precipitation" instead of "atmospheric deposition" if you only refer to water.

L174-180: The water fluxes in the text have different precision as in Fig. 2. This results in a 10% difference in the estimate of surface runoff of 1 or 1.1 km$^3$ yr$^{-1}$. Please use the same precision in the text as in Fig. 2.

L209-210: Equation 2 and 3: Please give information on how the regression equations were derived, by showing figures (in the supplement) or citing references if they were taken from publications. Were the data (BSi, POC and SPM) sampled from the Yellow River? If so, please state the assumption that the relation between BSi/POC and SPM in the other rivers is similar as that in the Yellow River.

L221: The area of Bohai Sea should be introduced along with the average depth already in the introduction.

L225-226: Is the DOC concentration in the coastal rainwater, reported by Willey et al. (2000), a global (coastal region) average value or from a certain coastal region? Is there a potential difference than the value in Bohai Sea?

L239: Equation 4: $\partial C/\partial C$ --> $\partial C/\partial z$

L254-255: "Sedimentation *rate* of BSi and OC … from the accumulation rate *of total mass* and the surface area." I don't understand what is "the surface area".

L290-291: First, they are not similar to me at all. Second, do you mean the DSi concentrations are fairly low in the northwestern part of the Bohai Sea *in the bottom water*? Because it doesn't apply to the surface water.

L302-303: Again, the distribution of BSi in surface water is not similar to that in bottom water to me.

L306: I guess you meant lower, not higher. And please state here and where else in this section that it's the *average* concentration of all stations.

L317-318: The *average* BSi content … a large variability in both BSi and TOC *among stations*.

L328: The estimated deposition flux of what?

L334: "are similar" this statement is not appropriate and not necessary, please rephrase it.

L350: "The estimated surface runoff fluxes are 2.1 Gmol yr$^{-1}$ for RSi and 6.4 Gmol yr$^{-1}$ for OC *according to Eq. (1)*."

Section 4.1: See General Comment 1.

Section 4.2: The number of sampled stations in fall is 2.5 times as many as in spring. This is just a question to the authors whether the different sample sizes of two seasons have been taken into account in the analysis and the conclusion drawn from it.

L415-417 and L422-423: The dominant output should be sedimentation (99%), not sedimentation (99%) and export (1%). The same for L422-423.

L435-436: The work by Treguer and De La Rocha (2013) can also be cited here.

L439-442: Whether the benthic flux of DSi plays a role in maintaining the primary production or not depends on the euphotic layer depth and vertical mixing level. Seasonal variation may exist also. Please elaborate on this point.

L445: Before saying "another way", the authors should probably state "your way" first, which means to move the last sentence in front of "Another way...".

L452: "*The ratio of BSi sedimentation flux to BSi production* (3.3%) exceeds the average value for the world ocean (2.6%)."

L454: "" → within the range of

L464: "Relative … and riverine Si input *of Bohai Sea* are similar to …." Please cite numbers here to demonstrate how similar they are.

L468: What are the "above studies"? Do you mean "(Liu et al., 2011)"? Please clarify here. In addition, this is only one study, please also modify the phrase "Previous studies" at the beginning of the paragraph accordingly.

L480-482: This sentence belongs to the paragraph above.

L482: "silica" refers usually to the diatom shell, which is BSi as in the manuscript. Maybe you meant DSi or RSi here?

L495: It seems to me a conflict between "annual" and "in July".

L496: "this year" can be misleading as 2016

L501: My suggestion: "This is confirmed by DOC increase in the Bohai Sea from 2.1 mg l$^{-1}$  (Zhang et al., 2006) to 2.6 …;"

L501-506: How much has the riverine input of OC changed? Can the increase of OC concentration in the Bohai Sea be attributed to increase of riverine input of OC?

L530-532: This sentence belongs to Introduction.

L532-534: The authors should indicate here that the 10% increase in PP comes from a calculation based on the relation between RSi and PP in year 2000-2008.

Section 5: It would be good to mention what the major uncertainty is in the budget estimate and to comment on the potential bias on the estimate due to the assumption of steady-state at the end of this section.

***Comments on figures and tables:***

Table 1: Main fluxes of reactive  *silicon* and …

I found it difficult to find information in this table. The most interesting numbers might be the Si and C fluxes from different sources, which should be kept in the last two columns for each source, as you did for $F_R$. It would also be nice to have a sum of all rivers for Si and C fluxes as an additional row below "Daling River". Wherever possible, the same terms should be kept in the same column, e.g. for water exchange ($F_E$), just let the third column empty and move the rest six columns to the right.

Table 2: Reactive  *silicon ...*

I would suggest to move "Spring" and "Fall" in the same row as "Surface" to avoid confusion, or to draw a horizontal line to separate two seasons. The same suggestion goes to Table 3 for "Surface sediment" and "Core sediment".

Figure 1: Please state the core sediment sampling stations in the legend.

Figure 3 and 4: They don't provide new information or systematic understanding, in my opinion. Therefore they can be removed. Also see the General Comment 1.

Figure 5: reactive  *silicon …*

This is a very nice figure! Maybe also state the unit in the legend?

**Reference**

Li et al., 2003, Distributions of inorganic nutrients in the bohai sea of china, Journal of Ocean University of Qingdao, Volume 2, Issue 1, pp 112-116

---

## Referee Comment (RC2) · Anonymous Referee #2 · 25 Mar 2016

The manuscript describes the results from two campaigns in the Bohai Sea, China and computes Si and C budgets for the coastal system. There were many measurements made, but also data from the literature was necessary to compute budgets. The methods section is not clear, so it is difficult to partly understand how the measurements were made and how many samples were collected. Although large spatial gradients were observed in water column Si and C concentrations, a 1D box model integrated over the coastal bay was used to calculate the budgets. The primary production of the system was then related to the water, sediment and nutrient inputs from riverine inputs.

I am not confident in the adequacy of the measurements or the budget produced. Details regarding the results and calculations in the manuscript are not well described.

The manuscript requires extensive revisions. However, once revised it will not make a significant contribution to the biogeosciences.

Specific comments

How many water samples were taken in Bohai Bay and where were they located?

Lines 66-76 This paragraph is difficult to understand. Both changes in the river and in the Sea are mixed together. They present the idea that river regulation of the Yellow River has changed, but do not mention what changes were made.

Line 93 Two campaigns for water column measurements are inadequate to produce meaningful mass balances.

Lines 103-105 Were the filters cleaned using this method or were the samples processed using this method?

Line 225 Was atmospheric deposition from a model or from measurements?

Lines 277-284 Using satellite remote sensing to calculate primary productivity in a coastal area with sediment inputs is difficult. Further, extrapolation of uptake rates from standard nutrient ratios is not sufficient.

Lines 368-369 If there was a large spring diatom bloom, couldn't the high bottom water column concentrations be due to settling?

Lines 480-482 The dissolved silica concentrations show not evidence for Si limitation of DIATOMS. However, even if the Bay becomes dissolved silica limited there are other algae that do not require Si, so I do not understand how the system is not limited by N and/or P.

There are an excessive number of references.

The BSi data in Table 3 has extremely low numbers in the sediments and surprisingly low standard deviations, especially when you compare it to the data in Figure 4 that has

large spatial variations in BSi. In addition, sediment BSi concentrations then to be relatively invariant with time, but there are large differences spatially in BSi concentrations between the spring and fall samplings. I find it difficult to believe these numbers.

---

## Author Comment (AC1) · 7 May 2016

Dear Reviewer,

Thank you very much for your attention and the useful comments and suggestions for improvement on our paper BG-2016-42. Based on the comments, we have made extensive modifications (point by point) of the original manuscript. We attach the revised manuscript with changes marked as PDF, as well as the document with our responses to your comments.

Kind regards, Jun Liu, Lex Bouwman, Jiaye Zang, Chenying Zhao, Xiaochen Liu, Zhigang Yu, Ran Xiangbin

[Figure]

Please also note the supplement to this comment:
http://www.biogeosciences-discuss.net/bg-2016-42/bg-2016-42-AC1-supplement.zip